# Analysis of Carbon Emission Reduction at the Port of Integrated Logistics: The Port of Shanghai Case Study

**Yilin Zeng, Xiang Yuan * and Bing Hou**

School of Economics and Management, Shanghai Maritime University, Shanghai 201306, China;
cengyilin@stu.shmtu.edu.cn (Y.Z.); 18249855363@163.com (B.H.)
* Correspondence: yuanx2109@163.com

**Abstract:** With the goal of achieving carbon neutrality in the shipping industry, the issue of sustainable port development is becoming more and more valued by the port authorities. The shipping industry requires more effective carbon emission reduction analysis frameworks. This paper takes China's Shanghai Port as the research object and analyzes it from the perspective of port-integrated logistics. Combined with the port data of Shanghai Port from 2008 to 2022, the principal component analysis gray correlation analysis model was used to screen the factors affecting the port's carbon emissions, and three calculation models for Shanghai Port's carbon emission sources were proposed. In addition, an expanded stochastic impact model based on the regression of population, affluence, and technology (STIRPAT) was constructed for the influencing factors of Shanghai Port's carbon dioxide emissions and combined with the method of ridge regression to further identify important influencing factors. At the same time, a gray neural network model was established to predict the carbon emissions of Shanghai Port from 2021 to 2030 and compare them with their real value. The conclusion shows that there is a close relationship between Shanghai Port carbon emissions and container throughput, throughput energy consumption, number of berths, total foreign trade import and export, and net profit attributable to the parent company. Gray neural network model data calculations show that the growth rate of Shanghai Port's carbon emissions will gradually slow down in the next ten years until the carbon peak is completed around 2033. The study can provide a reference for the sustainable development of other ports.

**Keywords:** sustainable port; carbon emissions forecast; integrated model; port-integrated logistics

## 1. Introduction

With the gradual development of the world economy, global environmental problems have become increasingly acute. In recent years, problems such as the greenhouse effect, melting glaciers, and the hole in the ozone layer have emerged one after another, bringing endless troubles to mankind. These ecological crises have attracted worldwide attention, and the control of carbon dioxide emissions has also become the focus of attention [1]. Under the new situation of deteriorating global environmental quality and an increasingly severe energy crisis, the "green industrial revolution" caused by global climate deterioration is being promoted all over the world, and more and more countries and regions have begun to realize the importance of green development and ecological environment protection [2]. It is of great significance to begin to actively deal with the issue of carbon emissions and introduce relevant control policies and low-carbon development strategies in order to achieve the goal of controlling carbon emissions. As the country with the largest carbon emissions in the world, China is determined to carry out energy conservation and emission reduction. In September 2020, the Chinese government put forward the "3060 Dual Carbon Goals". China will reach the peak of carbon emissions in 2030 and achieve carbon neutrality in 2060. The port of China's economic artery has become an important goal of China's energy conservation and emission reduction [3].

As an important node in the modern logistics supply chain, the port has generated huge economic benefits from its daily activities [4]. However, while the port brings economic benefits, its production activities will also cause a lot of pollution, including water pollution, noise pollution, air pollution, and so on. With increasing global trade, the port industry is also becoming more prosperous, but correspondingly, the port also produces a large amount of carbon dioxide. According to the IPCC's (United Nations Intergovernmental Panel on Climate Change) "Special Report on Global Warming of 1.5 °C" [5], the port industry accounts for about 3% of the total global greenhouse gas emissions. In 2011, the carbon emissions of the ports were 779.50 million tons, and they rose slowly. In 2013, the carbon emissions of the ports reached 838.33 million tons. After 2013, the carbon emission trend of the port has been stable and declining, and the carbon emission in 2018 was 810.89 million tons. Undoubtedly, as an important part of international shipping, taking appropriate measures to reduce port carbon emissions is of great significance to the achievement of the shipping industry's carbon emission reduction goals [6].

Shanghai is an important economic, transportation, technological, industrial, financial, and shipping center in China and one of the largest metropolitan areas in the world in terms of scale and area. The cargo throughput and container throughput of Shanghai Port rank first in the world, and it is a good riverside and seaside international port. Shanghai is also the location of China's first free trade zone, the "China (Shanghai) Pilot Free Trade Zone". The Yangtze River Delta urban agglomeration, composed of Shanghai, Jiangsu, Zhejiang, and Anhui, has become one of the six world-class urban agglomerations [7]. Shanghai attaches great importance to energy conservation and emission reduction. Recently it issued the "Shanghai 14th Five-Year Plan" Comprehensive Work Implementation Plan for Energy Conservation and Emission Reduction [8], which proposes that "By 2025, the energy consumption per unit of GDP will have been reduced by 14% compared with 2020, and the total energy consumption will be reasonably controlled." The emission reductions of key projects for the four major pollutants of nitrogen oxides (NOx), volatile organic compounds (VOCs), chemical oxygen demand (COD), and ammonia nitrogen (NH3-N) reached 13,000 tons, 9900 tons, 16,300 tons, and 1200 tons, respectively. The policy mechanism for energy conservation and emission reduction has become more sound; the energy utilization efficiency of key industries and the control level of major pollutant emissions have basically reached the international advanced level; the recycling industry and social system have basically formed; and the green transformation of economic and social development has achieved remarkable results. Shanghai is actively taking relevant measures to reduce carbon emissions.

In recent years, the issue of port carbon emission reduction has attracted the attention of many scholars. Most scholars have analyzed how to reduce port carbon emissions from the perspective of port energy conservation and emission reduction efficiency measurement. For example, using the super-efficient SBM model (an efficiency measurement model was improved on the basis of the traditional DEA model, which can exclude constraints whose efficiency value is less than 1.) to measure the energy conservation and emission reduction efficiency of the Bohai Rim port group to evaluate and explore the factors that affect port carbon emission reduction and give corresponding policy recommendations [9]. At the same time, some scholars have conducted research on the use of shore power in ports and have given a carbon emission calculation model [10]. In short, most scholars' research discussions are about the optimization of port operation and energy use schemes or how to provide alternative transportation solutions to achieve the purpose of reducing port carbon emissions. However, the current academic research on the multi-level driving factors that may affect carbon emissions is still insufficient. Most studies are limited to the study of carbon emissions in a single port and lack analysis at the level of the port's integrated logistics system. In addition, due to inaccurate forecasts for ports across China, there is also a shortfall in developing targeted strategies for low-carbon emissions in ports. On this basis, this paper proposes an innovative, comprehensive framework to analyze the carbon emission issue of port-integrated logistics systems and explore the potential driving

factors of carbon emissions. Taking Shanghai Port as an example, this paper proposes three calculation models of Shanghai Port's carbon emission sources, predicts carbon emissions, and further gives corresponding policy recommendations.

The rest of the paper is structured as follows: Section 2 presents the related work of this study; Section 3 outlines the methodology employed in the article; Section 4 states the data sources, conducts variable selection and empirical analysis, and discusses the results. Finally, Section 5 presents targeted policy recommendations, and Section 6 draws conclusions.

## 2. Literature Review

### 2.1. Research on Port Carbon Emission

Reducing greenhouse gas emissions and developing new emission reduction technologies are important measures to develop green and sustainable ports [11]. The current academic research on port carbon emission reduction can be roughly divided into two perspectives: macroscopic and microscopic [12].

From a macro perspective, Lingpeng Meng et al. [13] believe that with the increase in additional costs of government supervision and the reduction of incentives for port enterprises, the probability of the government's active regulation of carbon emission reduction will increase, and the probability of port enterprises' initiative to implement carbon emission reduction will increase accordingly. Increases over time, reducing the additional costs of passive mitigation and increasing opportunity costs. Likun Wang et al. [14] found that the carbon emissions of port container transportation are negatively correlated with the local GDP (Gross Domestic Product) and the number of port berths and positively correlated with the value of the local tertiary industry, road freight volume, and local and surrounding waterway freight volume. Lei Yang [15] and others believe, from the perspective of society, that when the carbon price and environmental concern are low or high, low-sulfur fuel oil should be used. Otherwise, shore power may be more attractive. Wang Shuang [9,16] and others found that environmental supervision is the main factor affecting the green energy saving efficiency of the Bohai Port Group, and technical factors also have a positive effect on energy saving and emission reduction in port groups. At the same time, the comprehensive economic strength has a positive effect on the improvement of the green energy efficiency of the port group.

At the microlevel, Xiaoyan Guo et al. [17] believe that transportation structure and fuel choice significantly affect network emission reductions. Yu Yao et al. [18] argue that port carbon emissions are strongly linked to port throughput, productivity, containerization, and intermodal transport. Ling Sun [10,19] believes that only a small number of coastal provinces and cities are suitable for using shore power, and they are limited by the berthing times of ships. However, this condition has nothing to do with the size of the ship but with the power generation ratio. Sheng-Long Kao [20] and others believe that the impact of shore power supply on carbon emission reduction is significantly greater than the speed limit policy. Yu-Chung Tsao et al. [21] found that the development of the dry port concept and intermodal transport can reduce the carbon cost of road transport. Lower speeds and onshore power availability can reduce local air pollution, by Hui-Huang Tai et al. [22].

In terms of emission reduction measures, Lei Yang and others took Shenzhen Port as an example and proposed methods such as improving loading and unloading efficiency, replacing heavy fuel oil with low-sulfur fuel oil, and shore power [15,23]. Bei Wang and others proposed to conduct a comprehensive study of port emission inventories and emission reduction technical measures and analyze them according to the actual situation of different ports [11,24]. Yu Yao et al. proposed that China's port authorities need to increase the proportion of containerization and develop multimodal transport; at the same time, under the new vision of clean energy and automation equipment, according to the optimization of port operation management, including peak shaving and intelligent management systems, the port authorities are responsible for updating energy use and energy efficiency to minimize the proportion of non-green energy consumption [18,25]. Sheng-Long Kao

et al. propose to balance the emissions improvement scenario by combining a new speed policy with a 50% shore power supply [20,26].

*2.2. Research on Carbon Emission Analysis Method*

The calculation and analysis of port carbon footprints are key steps in evaluating port carbon emissions. At present, many scholars in the academic circle use various methods to study the issue of port carbon emission reduction. Xiaoyan Guo et al. [17,27] developed a carbon emission estimation model for a hinterland-based container intermodal network. Taking Shanghai Port and the Yangtze River Delta (YRD) hinterland as examples, the patterns of well-to-wheel (WTW) and tank-to-wheel (TTW) over the past decade are estimated. Key drivers are identified through sensitivity analysis, and changes in carbon emissions under different carbon reduction policy scenarios are analyzed. Taking Shenzhen Port as an example, Lei Yang et al. [15,28] provided a method to measure the carbon emissions of the integrated logistics system of the port based on the comprehensive logistics perspective of the port. Likun Wang et al. [14,29] propose an easy-to-implement method for calculating $CO_2$ emissions from port container distribution (PCD) and study their spatial characteristics and drivers. Yu Yao et al. [18,30] proposed an integrated framework, combining population, affluence, and technology regression (STIRPAT), global Malmquist-Luenberger (GML), and multiple linear regression (MLR) random effects models to explore the drivers of carbon emissions from Chinese ports. Shumin Lin [31] proposed a mathematical model aimed at minimizing the sum of the total carbon emission cost and the total penalty cost to study the comprehensive optimization problem of the space allocation of tidal port berths, quayside cranes, and storage yards with channel capacity constraints under the carbon tax policy. Yao Yu et al. [12,32] proposed the Stochastic Effects of Population, Wealth, and Technology Regression (STIRPAT)-long short-term memory (LSTM)-autoregressive integrated moving average (ARIMAX) composite model with explanatory variables for estimating carbon emissions. Sheng-Long Kao et al. [20] combined the Automatic Identification System (AIS), Ship Emission Estimation Model (SEEM), Geographic Information System (GIS) mapping, and scenario simulation technology to create a Ship Emission Scenario Simulation Model (SESSM) for mapping and evaluating the current ship emissions. Xing Jiang [33] proposed an adaptive, dual-population, multi-objective genetic algorithm, NSGA-II-DP, to calculate ship channel scheduling and berth allocation problems. Hui-Huang Tai et al. [22] proposed an activity-based model to calculate ship exhaust emissions. Yuyan Zhou et al. [34] used the WRF-CMAQ model to estimate the impact of port-related source emissions on air quality. Wang Shuang et al. [9] used the super-efficiency SBM model to measure the energy-saving and emission-reduction efficiency of ports. In addition, based on the STIRPAT model, the influencing factors of energy savings and emission reduction efficiency were constructed for analysis.

In summary, only a few quantitative studies evaluate and predict the assessment and prediction of carbon emissions in ports from the perspective of port comprehensive logistics. In this study, a systematic combination method is adopted. Based on the carbon emissions data of Shanghai Port over the years, the carbon emissions volume of Shanghai Port is predicted and analyzed, and the problems existing in Shanghai Port's carbon emission reduction correspond to solutions.

## 3. Methodology

This paper takes Shanghai Port, located in East China, as the research object. First of all, by consulting the "Sustainable Development" reports issued by Shanghai Port Group over the years, the IPCC method is used to calculate the carbon emissions of Shanghai Port. Secondly, according to the existing research, the index of the influencing factors of carbon emissions is determined as a candidate set, and the principal component analysis method is used for noise reduction. Then, the statistical software SPSSPRO (Scientific Platform Serving for Statistics Professional, which is a new online data analysis platform that is different from the traditional client mode of SPSS and SAS.) uses gray relational analysis

to calculate the correlation degree of the candidate set after noise reduction and screen out and organize the factors that have an important impact on port carbon emissions. At the same time, the influence of key factors is quantified through the Stochastic Influence Expansion Model Based on Population, Wealth, and Technology Regression (STIRPAT) and multiple regression models, and the factors related to port carbon emissions are analyzed in detail. Finally, the gray prediction model-BP neural network model (GM(1,1)-BP neural network model) is used to predict the carbon emissions of Shanghai Port, analyze the carbon emission reduction problem of Shanghai Port, and give policy suggestions at the same time. The overall flow of the method in this paper is shown in Figure 1.

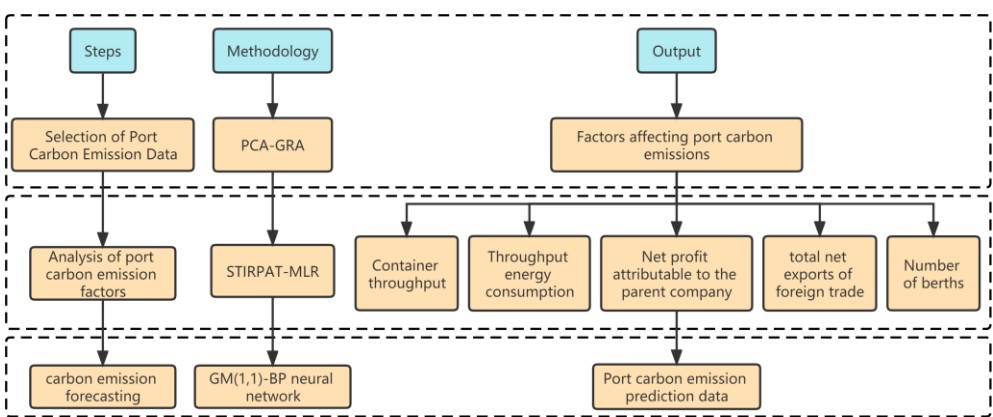

**Figure 1.** Flowchart of the proposed model.

### 3.1. Research on Port Integrated Logistics

Port-integrated logistics mainly provides warehousing and transshipment services with the port as a node and integrates various services, such as agency, processing, distribution, procurement logistics, finance, and information processing. These services are integrated into three centers: the logistics service center, the business service center, and the information service center, to provide users with multi-functional and comprehensive logistics services [35]. As shown in Figure 2.

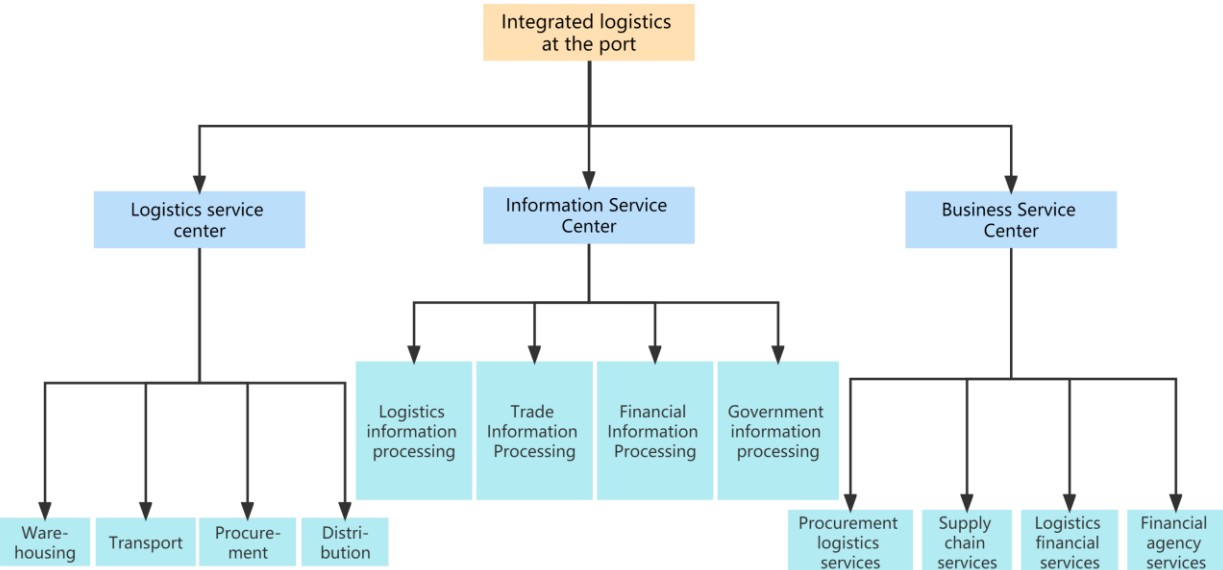

**Figure 2.** Schematic diagram of the integrated logistics system of the port.

The logistics service center is the main feature of comprehensive port logistics. Its functions are developed from the warehousing and transportation services in traditional

port logistics, and now they also include procurement, distribution, agency, processing, and other businesses. The purpose is to serve the personalized logistics needs of different customers.

The business service center means that in addition to traditional warehousing and transportation services, the port also provides customers with commercial and financial services such as agency, insurance, and banking, as well as modern logistics services such as procurement logistics, supply chain services, and logistics finance. For example, it could use its own resource advantages to develop procurement logistics or cooperate with other companies to establish a supply chain service platform.

The information service center has an important function that is different from traditional port logistics. Its main task is to process and feed back information on logistics, trade, finance, and government affairs, to provide customers with information services, and to realize the interaction of business flow, logistics flow, and information flow. For example, establish an information sharing database to improve the information management level of enterprises in the port. Port integrated logistics provides customers with multi-functional, integrated, and personalized integrated logistics services through these three centers to gain competitive advantages.

### 3.2. PCA-GRA

Principal Component Analysis (PCA) is an index screening method. Its principle is based on the idea of dimensionality reduction and realizes the method of integrating multiple scattered indicators into a small number of comprehensive indicators [36]. Gray relational analysis (GRA) is a multi-factor statistical analysis method. The basic idea is to judge whether the connection is close according to the similarity of the geometric shapes of the sequence curves. The closer the similarity of the corresponding shapes of the curves of different sequences is, the higher the correlation between sequences is, and vice versa [37]. The advantage of PCA-GRA (Principal Component Analysis-Grey Relation Analysis) is that it can greatly reduce the loss caused by information asymmetry and has low data requirements. According to the changing situation of two factors (direction, size, speed, etc.), you can judge the relationship between the two. The steps of PCA-GRA are as follows:

Step 1: Select the analysis index sequence and establish the reference sequence and comparison sequence. Among them, the reference sequence refers to the data sequence that reflects the characteristics of the system, and the comparison sequence refers to the data sequence composed of factors that affect the behavior of the system. The reference sequence is recorded as $X_0 = [x_0(1), x_0(2), \ldots, x_0(n)]$, and the comparison sequence is $X_i = [x_i(1), xi(2), \ldots, x_i(n)]$, $i = 1, 2, \ldots, m$. In this paper, the reference sequence is the carbon emissions of the Shanghai port, and the comparison sequence is the data set related to the carbon emissions of the Shanghai port found in the existing literature and research.

Step 2: Dimensionless processing of data sequences. The dimensions of the original data series are usually different, so the original data series should be dimensionless for the convenience of data comparison and analysis. The specific formula is as follows:

$$X_i'(k) = \frac{x_i}{x_i(1)} \left(k = 1, 2, \ldots n, \ i = 1, 2, \ldots n'\right) \tag{1}$$

The reference sequence is dimensionless, and the comparison sequence is dimensionless to obtain $X_0^t = [x_0'(1), x_0'(2), \ldots, x_0'(n)]$. Dimensionless comparison sequence to get $X_i'(1) = [x_i'(1), x_i'(2), \ldots, x_i'(n)]$, $i = 1, 2, \ldots, m$. k is one of the subsequence identifiers.

Step 3: Calculate the absolute value of the difference between the reference sequence and the comparison sequence, namely $|x_0(k) - x_i(k)|$, $(k = 1, 2, \ldots \ldots n, \ i = 1, 2, \ldots \ldots n')$, and find the maximum and minimum values in it, which are denoted as a and b, respectively.

Step 4: Calculate the correlation coefficient of each indicator in the comparison sequence and the reference sequence; the specific formula is as follows:

$$\gamma(x_0(k), x_i(k)) = \frac{a + \rho b}{|x_0(k) - x_i(k)| + \rho b} (k = 1, 2, \ldots \ldots n, \; i = 1, 2, \ldots \ldots n') \quad (2)$$

Among them is the resolution coefficient, which generally takes a value in the range of 0 to 1 and is usually selected.

Step 5: Calculate the correlation degree.

$$\gamma_{0i} = \gamma_{0i} \sum_{k=1}^{n} \gamma_{0i}(k), i = 1, 2, \ldots, m \quad (3)$$

In the formula, $\rho$ is the discriminant coefficient, and the smaller $\rho$ is, the higher the discriminant rate is. By default, $\rho$ is set to 0.5, the recognition effect is moderate, and the stability is good.

### 3.3. STIRPAT and MLR

Today, the STIRPAT model is widely used to solve the problem of peak carbon emissions. Compared with other models, the STIRPAT model more accurately measures the impact of social and economic factors on the environment. It examines the influence of the environment on the problem of peak carbon emissions [28]. Compared with the environment, it eliminates the impact of the same proportion of changes. At the same time, the STIRPAT model also has strong scalability, which can change with changes in actual problems. The basic form of the model is:

$$I = aP^b A^c T^d e \quad (4)$$

In this formula, I is the environmental pressure; P is the population factor; A is the wealth factor; T is the technical level; a is the model coefficient; e is the random error disturbance of the model; and modulus of elasticity in previous studies, due to the difficulty of obtaining data, few scholars have verified the impact of the port's net profit on its carbon emissions. And this paper expands the formula to include port factors, economic factors, technical factors, and other factors and incorporates the port's net profit attributable to the parent company into the calculation model, which is as follows:

$$I = aP^b A^c T^d G^f e \quad (5)$$

And under the premise of not affecting the stability of the original data, in order to reduce the volatility between the data, the logarithmic change processing is performed on both sides of the model equation at the same time, and (3) is obtained:

$$\ln I = \ln a + b\ln P + c\ln A + d\ln T + f\ln G + \ln e \quad (6)$$

Among them, G represents other factors, and f is the elastic coefficient of other factors. The extended STIRPAT model is used to measure the factors that affect the carbon emissions of the integrated logistics system of the Shanghai Port. Further, it is necessary to carry out multiple linear regressions on the established STIRPAT model to quantify the influence of the driving factors on Shanghai Port's carbon dioxide emissions.

### 3.4. GM (1,1)-BP Neural Network Model

The gray neural network prediction model is a combined model that combines the gray GM (1,1) model with the BP neural network model [38], and its construction steps are as follows:

Build a BP network model of the residual sequence. The residual of the original time series and the predicted value using the GM (1,1) model is recorded as $e^{(0)}(k)$. Let $\{e^{(0)}(k)\}(k = 1, 2, \cdots n)$ be the residual sequence and S represent the prediction order; that

is, use the information from $e^{(0)}(k-1), e^{(0)}(k-2), \cdots, e^{(0)}(k-S)$ to predict the value at time K. $e^{(0)}(k-1), e^{(0)}(k-2), \cdots, e^{(0)}(k-S)$ is used as the S input sample of the BP neural network, and the value of $e^{(0)}(k)$ is used as the predicted expected value.

Based on the trained BP neural network, predict the residual sequence $\{e^{(0)}(k)\}$ and determine $\{e^{(0)'}(k)\}$. Compute the new predicted values:

$$X^{(0)'}(k, 1) = X^{(0)'}(k) + e^{(0)'}(k, 1), (k = 1, 2, \cdots, n) \tag{7}$$

*3.5. Index Selection*

By reviewing the existing literature [1–30], we can find out the selection of factors affecting carbon emissions in the comprehensive logistics system of Shanghai Port, as shown in the following Table 1:

**Table 1.** Index selection.

| First-Level Indicators | Secondary Indicators | Variable | Unit |
|---|---|---|---|
| Port Factor | Container Cargo Throughput<br>Number of berths<br>Coastal Pier Length | $P_1$<br>$P_2$<br>$P_3$ | million TEU<br>individual<br>million meters |
| Economic Factors | Operating costs<br>Total assets<br>Total import and export of foreign trade<br>Net profit attributable to parent company<br>Gross Product of the Region | $E_1$<br>$E_2$<br>$E_3$<br>$E_4$<br>$E_5$ | 100 million yuan<br>100 million yuan<br>One hundred million U.S. dollars<br>100 million yuan<br>100 million yuan |
| Technical Factor | Throughput energy consumption | $T_1$ | tons/10,000 tons |
| Other Factor | Number of employees<br>Transhipment | $G_1$<br>$G_2$ | individual<br>none |

## 4. Empirical Analysis

### 4.1. Introduction of Port of Shanghai

Shanghai Port is the main gateway to China's international trade and the world's largest container shipping port. Shanghai Port is located in the middle of the east coast of mainland China, at the intersection of the "T"-shaped water transportation network formed by the "golden waterway" Yangtze River and coastal transportation channels. Anhui River and Taihu Lake water systems. The highway and railway networks criss-cross, the collection and distribution channels are smooth, the geographical location is important, the natural conditions are superior, and the hinterland economy is developed [39]. According to the "14th Five-Year Plan for the Construction of Shanghai International Shipping Center", the container throughput target of Shanghai Port will reach more than 47 million TEUs in 2025. In 2021, the annual container throughput of Shanghai Port will reach 47.025 million TEUs, and the second-placed Singapore Port will have an annual container throughput of nearly 10 million TEUs, further widening the gap. At the same time, Shanghai has steadily entered the top three in the ranking of international shipping centers. The main port area is distributed along the Huangpu River. It is composed of the port area on the south bank of the Yangtze River Estuary, the port area on the north bank of Hangzhou Bay, the Huangpu River port area, and the Yangshan deep-water port area Main portals and windows in Figure 3.

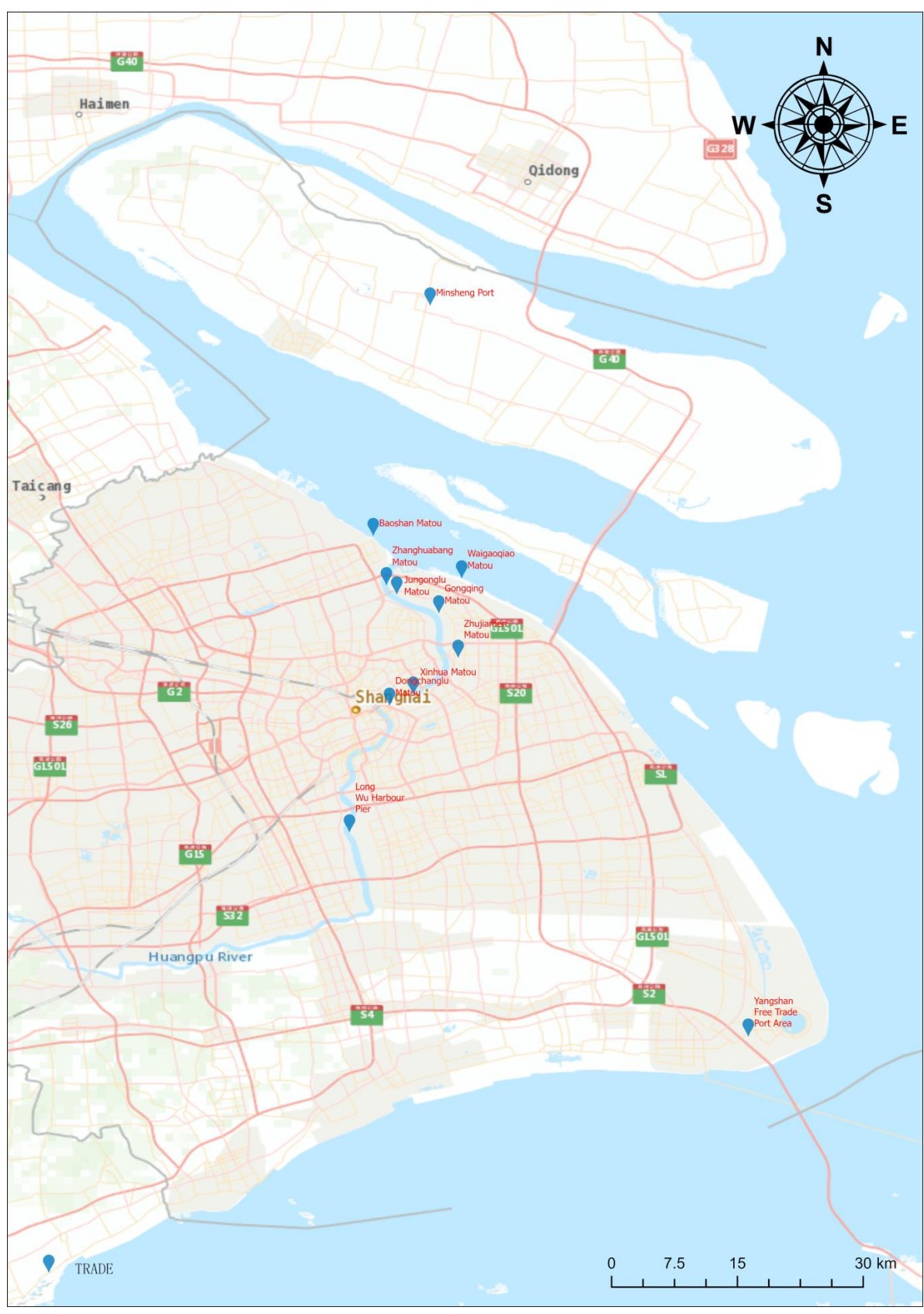

**Figure 3.** Map of Shanghai Port.

### 4.2. Analysis of Carbon Emission Sources in Port Integrated Logistics System

The carbon emission of the port integrated logistics system refers to the total emission of greenhouse gases produced by each link of the port integrated logistics system within a certain period of time. Due to the availability of data and the operability of analysis, this paper mainly measures the carbon dioxide emissions in the port's integrated logistics system emissions. The carbon emissions of the port integrated logistics system come from the services provided by the three types of functions of the system, including the logistics service center, information service center, and business service center. In the daily operation of the port, these services will directly or indirectly generate carbon emissions. By combining past literature and research [15–20], this paper mainly measures the carbon emissions of the port integrated logistics system from three aspects: port facilities, means of transportation, and material consumption [40]:

#### 4.2.1. Carbon Emissions Source Analysis in Port Facilities

The main source of carbon emissions in port facilities is the heavy equipment required for port services, including equipment for loading and unloading, handling, stacking operations, and storage at docks and yards. Which is necessary equipment for port services, usually including some equipment, such as shoreside container cranes, container forklift trucks, and terminal tractors. This article selects one of the shore container cranes for analysis:

$$Q_a^c = \begin{cases} g_e P_i^c C_i^{im} t_a^{im} & a \in A_{im} \\ g_e P_i^c C_i^{ex} t_a^{ex} & a \in A_{ex} \\ g_e P_i^c \left( C_i^{im} t_a^{im} + C_i^{ex} t_a^{ex} \right) & a \in A_{tr} \end{cases} \tag{8}$$

Among them, $Q_a^c$ is the carbon emission of loading and unloading container a by the shore container crane; $P_i^c$ is the rated power of the quayside container crane i in working condition; $C_i^{im}$ and $C_i^{ex}$ are the carbon emission intensity of the quay container crane i when loading and unloading the import container and export container, respectively; $t_a^{im}$ and $t_a^{ex}$ are the time spent by the quay container crane when loading and unloading the container a, respectively; and $A_{im}$, $A_{ex}$ and $A_{tr}$ are the collections of import boxes, export boxes, and transfer boxes, respectively. In the formula, it can be divided into three cases. The first case is for the import container $a \in A_{im}$, and $Q_a^c$ is the carbon emission $g_e P_i^c C_i^{im} t_a^{im}$ of the unloading operation of the shore container crane; the second case is for the export container $a \in A_{ex}$, and $Q_a^c$ is the shore container The carbon emission $g_e P_i^c C_i^{ex} t_a^{ex}$ of the crane; the third case is for the transfer box $a \in A_{tr}$, and $Q_a^c$ is composed of two parts: the carbon emission $g_e P_i^c C_i^{im} t_a^{im}$ of the unloading operation of the container crane on the shore, and the carbon emission of the $g_e P_i^c C_i^{ex} t_a^{ex}$ loading operation.

#### 4.2.2. Carbon Emissions Source Analysis in Transportation

The carbon emission sources in Shanghai port transportation vehicles mainly come from transportation vehicles, ships, and other transportation vehicles. This paper selects the ships arriving at the port for carbon emission analysis. The process of ships entering and leaving the port area can be divided into three stages: entering the port, berthing, and leaving the port. During the berthing phase, the operation of the auxiliary engine and boiler is usually maintained through fuel consumption, and if the ship is connected to shore power, the operation of the auxiliary engine is maintained through electrical energy. In the stage of entering and leaving the port, the ship is in a state of maneuvering, and the operation of the main engine, auxiliary engine, and boiler in a low-speed state is mainly maintained through fuel consumption. Based on the above analysis, the carbon emission calculation formula for the ship entering and leaving the port can be obtained:

$$Q_a^s = \begin{cases} g_f \left( P_a^s C_i^s t_a^s + P_a^b C_i^b t_a^b + Q_a^o \right) & a \in A_f \\ g_f (P_a^s C_i^s t_a^s + Q_a^o) + g_e P_a^e t_a^b & a \in A_e \end{cases} \tag{9}$$

Among them, $Q_a^s$ is the carbon emission of ship a entering and leaving the port; $g_f$ and $g_e$ are the carbon emission factors of fuel consumption and electricity consumption, respectively; $P_a^s$ is the main engine, auxiliary engine, and the average fuel oil consumption rate of the boiler; $P_a^b$ is the average fuel oil consumption rate of the ship a when the auxiliary engine and the boiler are in normal operation during the berthing phase; $P_a^e$ is the power consumption of the ship after using the shore power device; $C_i^s$ and $C_i^b$ are, respectively, the carbon emission intensity of ship a when it enters, leaves, and berths in state i; $t_a^s$ and $t_a^b$ are the time spent by ship a in the port entry and exit stage, and the length of berth; $Q_a^o$ is the fuel consumption of ship a due to auxiliary operation tools, such as tugboats when entering and leaving the port; $A_f$ and $A_e$ represent the situation that ship a does not use shore power after berthing, and the situation that ship a uses shore power after berthing. The formula is divided into two cases. In the first case, for a ship berthing without using shore power, $Q_a^s$ is composed of three parts: the carbon emission $P_a^s C_i^s t_a^s g_f$ of the ship's maneuvering state, the carbon emission $P_a^b C_i^b t_a^b g_f$ of the berthing state, and the carbon emission $Q_a^o g_f$ of other operations. Composition: the second case is that for ships using shore power, $Q_a^s$ is composed of three parts: carbon emissions $P_a^s C_i^s t_a^s g_f$ in the ship's maneuvering state, carbon emissions $Q_a^o g_f$ in other operations, and carbon emissions $g_e P_a^e t_a^b$ in berthing.

### 4.2.3. Analysis of Carbon Emission Sources in Material Consumption

In the daily operation of Shanghai Port, a large amount of material will be consumed. Ports and related enterprises need to consume a lot of paper documents, paper packaging, and materials for distribution and processing when various process handovers are provided. The carbon emissions from this part of material consumption are mainly due to the waste and recycling of materials. The calculation method proposed in this paper is as follows:

$$Q_w^s = \sum_i C_i^s W_i^s \tag{10}$$

Among them, $Q_w^s$ is the carbon emissions generated by the materials consumed in Shanghai Port's daily operations. $C_i^s$ is the amount of carbon dioxide generated by consuming 1 kg of material i. $W_i^s$ is the amount of material i. It can be seen from the formula that the carbon dioxide generated by the material consumption during the operation of the Shanghai Port in Shanghai Port is summed up by the carbon dioxide emissions generated by each kind of material.

### 4.3. Data Sources and Calculation Methods

This paper selects Shanghai Port, located in East China, as the research object. Considering data availability and quantitative requirements, the indicators and data for 2008–2022 required for this study are mainly obtained from the following sources: Construct a port carbon footprint calculation model, and calculate the corresponding total carbon footprint. this paper still refers directly to the "IPCC Carbon Emission Calculation Guidelines (2006)" for carbon emission factor default value data. The energy consumption of port throughput, container cargo throughput, water-to-water transfer rate, and the number of employees are all from the "Sustainable Development Report" issued by Shanghai Port Group over the years. The data on Shanghai Port's operating costs, total assets, and net profit attributable to the parent company are all from the "SIPG Financial Report" issued by Shanghai Port Group. The length of coastal wharves, the number of berths, the GDP of Shanghai, and the total amount of foreign trade imports and exports are derived from the Shanghai Statistical Yearbook and Shanghai Port Yearbook over the years.

### 4.4. Analysis of Carbon Emissions in Shanghai Port over the Years

A large part of carbon emissions in Shanghai Port is that three types of power carbon emissions, fuel oil carbon emissions, and diesel carbon emissions in the daily operations of ports were analyzed by SPSSPRO, and the data of electric carbon emissions, fuel

oil carbon emissions and diesel carbon emissions data in Shanghai. There is shown in Figures 4 and 5 below:

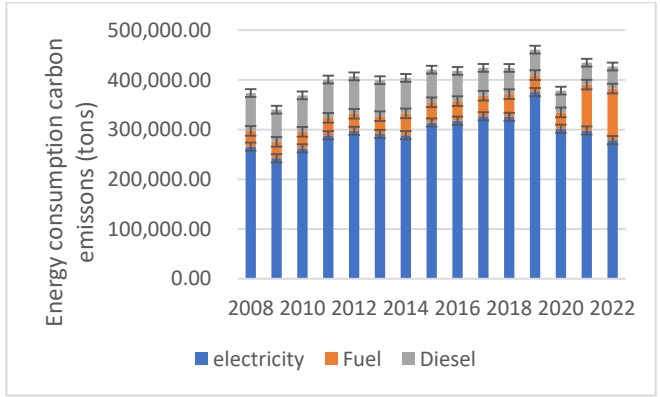

**Figure 4.** Three kinds of energy carbon emissions map.

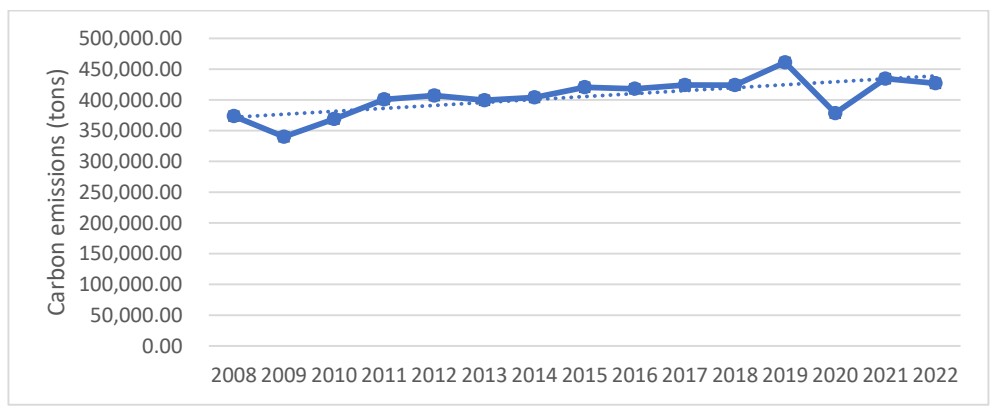

**Figure 5.** Shanghai Port Energy Carbon Emissions Map.

It can be seen from Figures 4 and 5 that from 2008 to 2022, the carbon emissions of Shanghai Port showed an overall growth trend. From 950,100 tons in 2008 to 1.2295 million tons in 2022, the average annual growth rate is 2%. Due to the impact of the new crown epidemic in 2020, the total carbon emissions of Shanghai Port will show a trough trend. From 950,100 tons in 2008 to 1.2295 million tons in 2022, the average annual growth rate is 2%. Due to the impact of the new crown epidemic in 2020, the total carbon emissions of Shanghai Port will show a trough. After the epidemic situation improves in 2021, the carbon emissions of Shanghai Port will pick up. Among the electricity carbon emissions, fuel oil carbon emissions, and diesel carbon emissions analyzed in this paper, from 2009 to 2019, the proportion of electricity carbon emissions has increased year by year, from 71.13% in 2009. To 81.46% in 2019. the proportion of Shanghai Port's carbon emissions to carbon emissions has increased by 10.33% in the past ten years; the proportion of fuel oil carbon emissions and diesel carbon emissions has decreased year by year, from 8.99% and 19.88% in 2009 to 2019 to 7.42% and 11.12% of the previous year, and the proportion of fuel oil carbon emissions and diesel carbon emissions in Shanghai Port has decreased by 1.57% and 8.66% in the past ten years; the proportion of fuel oil carbon emissions and diesel carbon emissions has decreased, which is in stark contrast to the year-on-year increase in the proportion of carbon emissions from electric energy, indicating that Shanghai Port Group is committed to using clean energy, reducing carbon emissions in Shanghai Port, and contributing to energy conservation and emission reduction. At the same time, it is also observed that from 2020 to 2022, the proportion of carbon emissions from electric power energy will drop from 79.73% to 68.63%, while the proportion of carbon emissions from fuel oil and diesel will increase accordingly. This is because since 2020, due to the fact that port

production activities have not been greatly affected, the volume of my country's import and export trade has increased significantly, which in turn has generated strong shipping demand [41]. Shanghai Port has increased fuel consumption when the original power facility scale cannot be expanded in a short period of time. The proportion of consumption of oil and diesel to meet the fast-growing shipping demand.

### 4.5. Identification of Carbon Emission Influencing Factors Based on PCA-GRA

Since the selected data samples are composed of 12 different index (The 11 indicators are in Table 1, and one variable is port carbon emissions.) feature variables and different index features have different dimensions and dimensional units, in order to eliminate the dimensional influence between different indicators and make different data indicators comparable at the same time without affecting the results of data analysis, this paper normalizes the data used and then performs principal component analysis and noise reduction processing through SPSSPRO. Before PCA, KMO and Bartlett sphericity tests were performed to determine the validity of the data [42]. The results are shown in Table 2 below. The KMO measurement value is 0.751 > 0.6, which shows that there is a correlation between the item variables, which meets the requirements of principal component analysis. The significance rate for Bartlett's test of sphericity for the chi-square statistic is less than 0.010. Therefore, the sample data is suitable for PCA.

**Table 2.** KMO and Bartlett's test.

| KMO and Bartlett's Test | | |
| --- | --- | --- |
| Kaiser-Meyer-Olkin Measure of Sampling Adequacy | | 0.751 |
| Bartlett's Test of Sphericity | Approx. Chi-square | 335.464 |
| | df | 55 |
| | Sig | 0.000 *** |

Note: *** indicate that the regression coefficients are significant at the 1% levels, respectively; t values are in brackets.

An analysis of variance was performed on dimensionless data. In the variance explanation table, when the principal component is 2, the characteristic root of the total variance explanation is lower than 1, so a total of 1 principal component is extracted, and the contribution rate of the variable explanation reaches 90.377 in Figure 6.

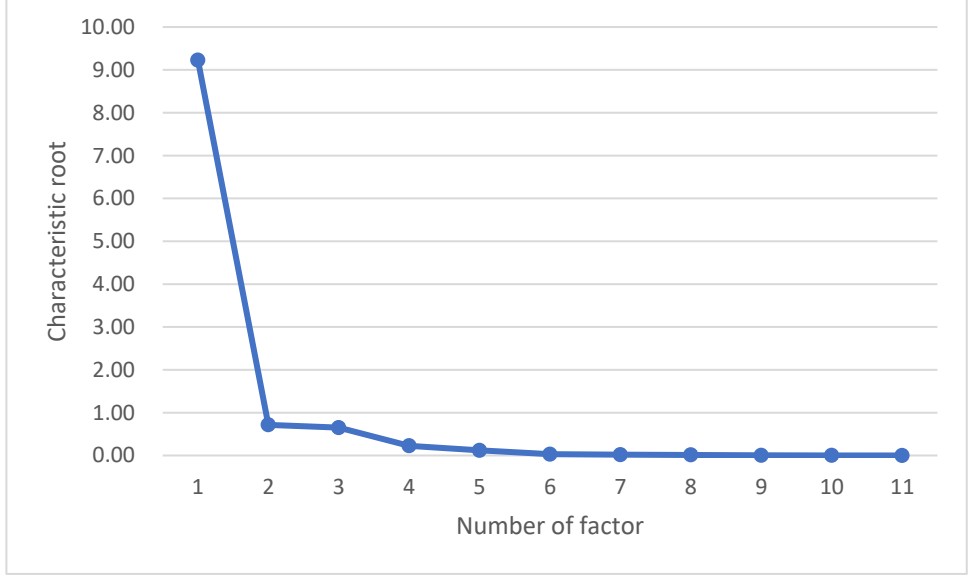

**Figure 6.** Factor Scree Plot.

Set the values to 0.094, eliminate the length of the coastal terminal and operating costs of the coastal terminal, and select the indicators of the container cargo throughput, the water rotation rate of water, the total value of the region, the total assets, the number of employees, the total foreign trade import and export amount, the number of berths, and the net profit of the mother-in-law and throughput energy consumption of these nine variables.

In this paper, the GRA method is used to further screen multiple characteristic variables. The correlation coefficient represents the subsequence $P_1$ (container cargo throughput), $P_2$ (number of berths), E2 (total assets), $E_3$ (total foreign trade import and export), $E_4$ (represented by parent net profit), $E_5$ (gross product of the region), $T_1$ (throughput energy consumption), $G_1$ (number of employees), and $G_2$ (water transfer rate) in relation to the value of the degree of correlation with the corresponding dimension of the parent sequence (the larger the number, representing stronger correlation), the gray correlation analysis diagram is as follows.

First, select the highest correlation factor in each module, and then set the threshold to $\alpha = 0.60$. In other variables, the factor will also be included in the final model, and the results are shown in Figure 7. According to existing research [9], screen and exclude G1 (number of employees). Through a comprehensive analysis of the impact of the carbon emissions of various indicators on the comprehensive logistics system of Shanghai and Port, this article finally chose the container cargo throughput (P1), the number of berths (P2), the total foreign trade import and export (E3), and the net profit of the mother (E4). The six factors of throughput energy consumption (T1) and water rotation rate (G2) were calculated.

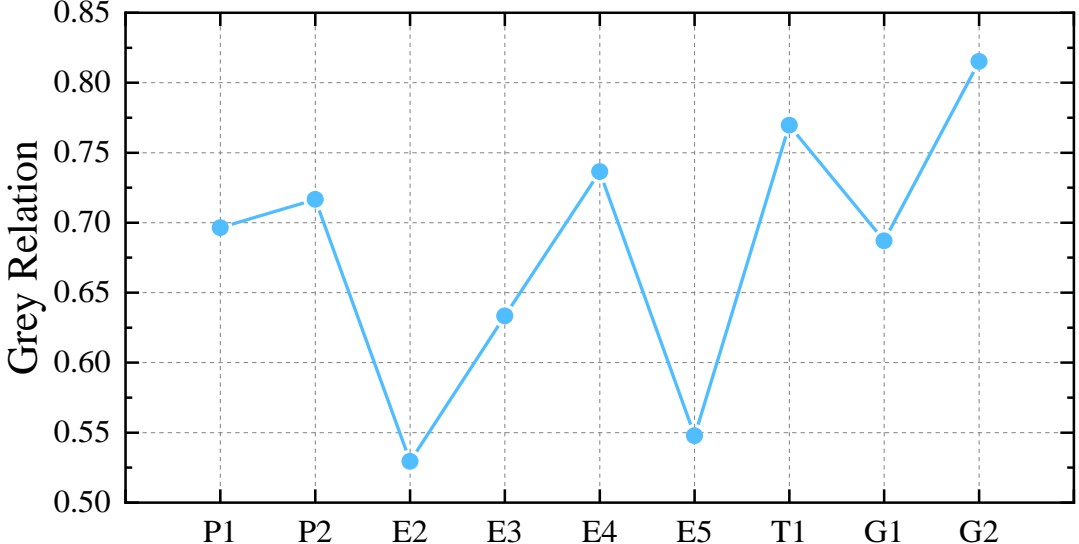

**Figure 7.** Shanghai Port Carbon Emissions Sequence Chart.

*4.6. Analysis of Improved STIRPAT Model*

According to the six variables previously selected through gray correlation analysis as explanatory variables (container cargo throughput, number of berths, total foreign trade import and export, net profit attributable to the parent company, and throughput energy consumption), Shanghai Port energy carbon emission is the explained variable. A model was built and screened to ensure the accuracy of the empirical analysis. Using SPSSPRO to conduct preliminary regression analysis, the results are shown in Table 3 below:

**Table 3.** Results of linear regression analysis.

| Variable | Non-Standardized Coefficient | | Standardization Coefficient | t | p | VIF | $R^2$ | Adjust $R^2$ | F |
|---|---|---|---|---|---|---|---|---|---|
| | B | Standard Error | Beta | | | | | | |
| Regression equation constant term | 0.9 | 0 | - | 12,402,841,551.451 | 0.000 *** | - | | | F = 9.636633925398487 $\times 10^{22}$ p = 0.000 *** |
| $lnP_1$ | 1 | 0 | 2.238 | 120,138,107,535.799 | 0.000 *** | 200.715 | 1 | 1 | |
| $lnT_1$ | 1 | 0 | 1.493 | 177,883,342,981.064 | 0.000 *** | 40.748 | | | |
| $lnE_4$ | 0 | 0 | 0 | 0.05 | 0.961 | 28.495 | | | |
| $lnP_2$ | 0 | 0 | 0 | 0.076 | 0.942 | 4.204 | | | |
| $lnE_3$ | 0 | 0 | 0 | −0.096 | 0.926 | 14.966 | | | |
| $lnG_2$ | 0 | 0 | 0 | 0.022 | 0.983 | 7.405 | | | |
| Due to variables: $CO_2$ emissions (ton) | | | | | | | | | |

Note: *** represent the significant level of 1%.

It can be seen from Table 3 that the analysis of the results of the F test can be obtained, the significance the *p* value is 0.000 ***, the level is significant, and the null hypothesis of a regression coefficient of 0 is rejected, so the model basically meets the requirements. However, among the independent variables, only the *p* values of container cargo throughput and throughput energy intensity are less than 5%, and the *p* values of the other four variables are all greater than 0.9, which is not significant in the model test. At the same time, it can be observed that the VIF values of container cargo throughput and throughput energy consumption intensity are 200.715 and 40.748, respectively. both of these values are >5, indicating that there is serious multicollinearity in the independent variables. Therefore, the coefficients fitted by OLS regression cannot be guaranteed to be accurate. When there is serious multicollinearity among the variables in the multiple linear regression equation, the variance and standard error of the OLS estimator will be relatively large, and the significance test will not pass, so the coefficients of OLS regression fitting cannot be guaranteed to be accurate. If OLS regression is used to analyze the influencing factors of port carbon emissions, this will lead to an inaccurate factor analysis and wrong conclusions. Therefore, OLS regression cannot be used here to analyze the influencing factors of port carbon emissions.

*4.7. Ridge Regression Results and Discussion*

In order to avoid the multicollinearity problem that may exist due to small independent variable samples, this paper adopts an improved least squares estimation method—ridge regression analysis—for data analysis and uses the ridge regression operation function in SPSSPRO to obtain different K values [43]. Standardized regression coefficients Through the ridge trace diagram, determine the K value. The selection principle of K value is that the minimum K value is reached when the standardized regression coefficient of each independent variable tends to be stable. In general, the smaller the K value, the smaller the deviation. According to the results of ridge regression analysis, when K = 0.172, the coefficient is gradually stable, R2 is 0.904, the F test passes the test at a significance level of 1%, and the fitting degree is good. Therefore, the parameter 0.172 is selected for regression, and the results are shown in Table 4 below:

**Table 4.** Results of Ridge Regression Analysis.

| K = 0.172 | Non-Standardized Coefficient | | Standardization Coefficient | t | $p$ | $R^2$ | Adjust $R^2$ | F |
|---|---|---|---|---|---|---|---|---|
| | B | Standard Error | Beta | | | | | |
| regression equation constant term | 10.478 | 0.45 | - | 23.269 | 0.000 *** | | | |
| $lnP_1$ | 0.084 | 0.016 | 0.187 | 5.099 | 0.001 *** | 0.938 | 0.904 | 27.41 (0.000 ***) |
| $lnT_1$ | 0.183 | 0.054 | 0.273 | 3.372 | 0.008 *** | | | |
| $lnE_4$ | 0.095 | 0.016 | 0.495 | 5.816 | 0.000 *** | | | |
| $lnE_3$ | 0.108 | 0.035 | 0.266 | 3.08 | 0.013 ** | | | |
| $lnP_2$ | 0.123 | 0.053 | 0.208 | 2.343 | 0.044 ** | | | |
| Due to variables: $CO_2$ emissions (ton) | | | | | | | | |

Note: ***and ** represent the significant level of 1% and 5%, respectively.

The regression coefficient of ridge regression is the elasticity coefficient. Based on the above analysis, the final linear model is:

$$lnI = 10.478 + 0.084 \times lnP_1 + 0.183 \times lnP_2 + 0.095 \times lnE_4 + 0.108 \times lnE_3 + 0.123 \times lnT_1 \qquad (11)$$

It can be seen from the formula that container cargo throughput, throughput energy consumption, net profit attributable to the parent company, total foreign trade import and export, and the number of berths have a positive impact on Shanghai Port's carbon emissions. It can be seen from the formula that the change percentages of port carbon emissions caused by a 1% change in variables from small to large are: container cargo throughput, 0.084%; net profit attributable to the parent company, 0.095%; total foreign trade import and export, 0.108%; throughput energy consumption, 0.123%; and number of berths, 0.183%.

*4.8. Prediction and Analysis of Carbon Emissions in Shanghai Port Integrated Logistics System*

The gray prediction model can predict irregular time series [44], which is in line with the data trends of Shanghai Port's carbon emissions, container cargo throughput, number of berths, total foreign trade import and export, net profit attributable to the parent company, and throughput energy consumption over the years.

Use the gray prediction method to suggest a GM (1,1) prediction model; record $x^{(0)}(k)$ ($k = 1, 2, \cdots, 20$). Taking the container cargo throughput, number of berths, total foreign trade import and export, net profit attributable to the parent company, and throughput energy consumption from 2008 to 2020 as the benchmark values, value is predicted the five variables and carbon emissions of Shanghai Port will be adjusted in 2021 to 2030.

After using the gray prediction model to predict that the carbon emissions of Shanghai Port in the next 10 years will show an increasing trend year by year, the neural network model trend extrapolation method is used to further improve the prediction results and predict the carbon emissions of Shanghai Port in the next 10 years. The BP neural network is a multi-level feedback network, which is a kind of fuzzy and uncertain neuron network that can carry out self-organization and self-learning. Since the BP artificial neural network has a thinking process similar to the human brain, it can simulate the human brain for continuous learning and training, so as to solve some problems with ambiguity and uncertainty [45]. Therefore, the artificial neural network is used to analyze the carbon emission samples of Shanghai Port, and through the learning, identification, and evaluation of new samples, the weight of each factor affecting the carbon emission of Shanghai Port is fully considered so as to predict the carbon emission of Shanghai Port in the next 10 years. Through MATLAB software (MATLAB is a commercial mathematics software program produced by Mathworks.), the historical data of Shanghai Port from 2008 to 2020 can be used to predict emissions in the next ten years. The predicted results are as follows in Table 5:

**Table 5.** Prediction Results of Carbon Emissions Based on Gray Neural Network.

| Years | Carbon Emissions (ton) |
|---|---|
| 2021 | 49,698,654.85 |
| 2022 | 49,728,648.53 |
| 2023 | 49,758,281.33 |
| 2024 | 49,785,596.70 |
| 2025 | 49,808,899.45 |
| 2026 | 49,827,299.75 |
| 2027 | 49,840,884.41 |
| 2028 | 49,850,429.03 |
| 2029 | 49,856,907.70 |
| 2030 | 49,861,152.41 |

Compare the obtained Shanghai Port carbon emissions results from 2020 to 2030 with the original data, as shown in Figure 8. Through the analysis of the fitting results of the model, it can be seen that the data curve fitting degree of the predicted curve and the actual value is high, and the data value is very close. The input layer in the model has 5 neurons, and each neuron corresponds to an impact factor (container cargo throughput, number of berths, total foreign trade import and export, net profit attributable to the parent company, and throughput energy consumption), and the input is these impact factors, which are the values after data normalization. The output layer can only have one neuron, which is the carbon emissions of Shanghai Port. Therefore, the established BP neural network model can accurately predict the future 10 years of Shanghai Port carbon emissions. Using the model to predict the carbon emissions of Shanghai Port from 2021 to 2030, they are 49,698,654.85 tons, 49,675,333.29 tons, 49,677,713.98 tons, 49,678,628.27 tons, 49,678,909.08 tons, 49,678,975.47 tons, 49,678,987.01 tons, 4,967,898 tons, 8.41 tons, 49,678,988.52 tons, and 49,861,152.41 tons. Figure 8 shows the change trend of the actual value, historical fitting value, and future forecast value of Shanghai Port's carbon emissions. Through the GM(1,1)-BP neural network model, the change in Shanghai Port's carbon emissions from 2021 to 2030 can be concluded: in the next 10 years, the carbon emissions of Shanghai Port will first increase and then decrease year by year (Figure 9). That is, the graph presents a relatively obvious inverted U shape, and based on the existing Shanghai Port carbon emission data, it can be obtained that Shanghai Port's carbon emissions will reach their maximum in 2033, and that 2033 is the peak moment of Shanghai Port's carbon emissions. The prediction results are also in line with the "3060" strategy proposed by the country.

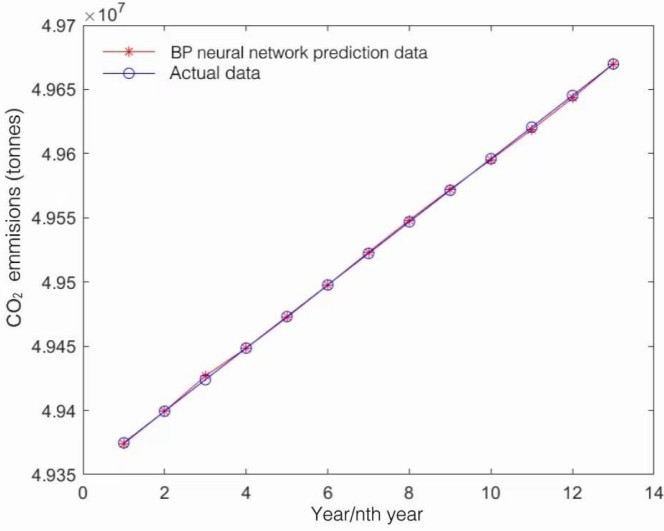

**Figure 8.** Gray Neural Network Training Diagram.

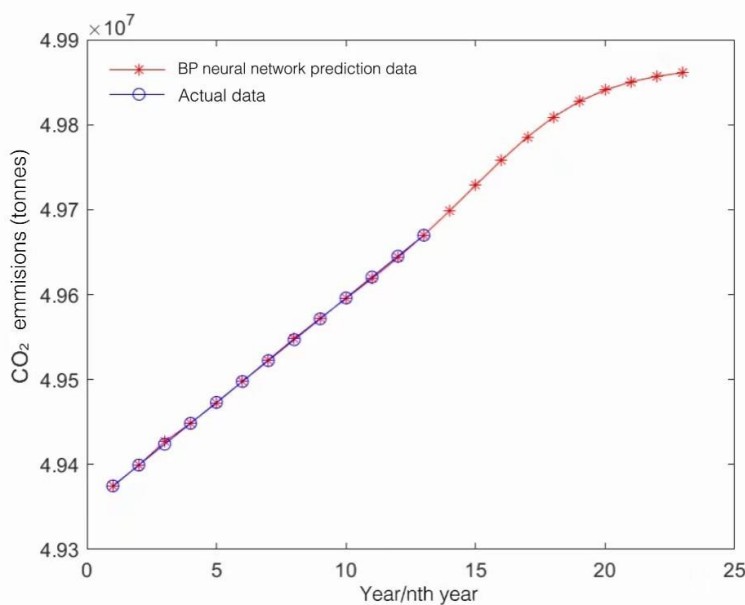

**Figure 9.** Gray Neural Network Carbon Emissions Forecast Map.

## 5. Strategies for Reducing Carbon Emissions in Port

### 5.1. Establish an Energy Forecasting and Early Warning Mechanism and an Emergency Response Mechanism to Meet the Needs of Trade Fluctuations

In 2020, Shanghai Port has increased the proportion of fuel and diesel consumption because electricity and energy consumption cannot meet the surge in shipping demand. Therefore, Shanghai Port needs to establish and improve the energy security reserve and purchase system to effectively control energy. Consumption and use, improve energy forecasting, early warning, and emergency response mechanisms, enhance energy security and emergency response capabilities, and build energy consumption emergency plans when trade demand fluctuates [11], so as to improve green energy security capabilities, ensure a stable, reliable, and effective supply of green energy, and empower green energy [46]. The construction of ports and green shipping have gained momentum.

### 5.2. Vigorously Promote the Construction of Green Ports and Improve Energy Efficiency

Ports can ensure the quality of equipment and reasonably control it through centralized bidding and procurement of equipment, reduce investment, and optimize resource allocation [47]. Compared with traditional fossil energy sources, such as coal and oil, port equipment uses fossil-free fuels, such as hydrogen or biofuels, or uses natural renewable energy sources, such as solar energy, wind energy, and hydropower, to generate electricity [48]. At present, electrification is a prominent trend in the port industry. The construction and use of new energy-saving and emission-reduction tools, such as photovoltaic power generation, "new energy vehicle" tire cranes, and automated heavy-box rail cranes, have a positive effect on protecting the ecological environment and will endow green ports, as green Shipping construction has greater momentum [49]. Ports can actively connect with shipping companies, carry out cooperation with leading shore power equipment manufacturing companies to carry out technical research, and promote the advancement of shore power technology and the improvement of safety performance and power connection success rates.

### 5.3. Optimize Port Operation Management

Shanghai Port Group can promote the conservation of internal resources and reduce the generation and discharge of pollutants by promoting green offices [50]. On the one hand, in terms of supplier management, suppliers are screened from various aspects, such as management systems, procurement processes, and environmental protection indica-

tors, and suppliers are required to provide high-quality and reliable equipment. priority principle [51]. On the other hand, we can compile an inventory of air pollution emissions, energy consumption, and carbon emissions and propose an index system for systematically evaluating the construction and development level of green ports.

*5.4. Improve Berth Utilization and Reduce Loading and Unloading Time*

According to the above regression results, it can be seen that the number of berths is directly proportional to the carbon emissions of Shanghai Port. However, the number of berths is generally difficult to change. At the same time, the increase and decrease of berths will also cost a lot of manpower, material, and financial resources [52]. Therefore, efficiency can be improved by increasing the utilization rate of berths and reducing loading and unloading times, so as to control the number of berths and promote green port development. On the one hand, by improving the service capacity and level of Shanghai Port, we can improve the utilization efficiency of berths, improve the channel conditions, and ensure that ships can enter and leave the port smoothly; on the other hand, we can promote technological progress, promote the progress of terminals and berths, and reduce comprehensive energy consumption and pollution emissions.

## 6. Conclusions

Since the implementation of China's "3060" strategy, many enterprises in China are committed to reducing carbon emissions generated in their daily business activities and making themselves more sustainable. This study proposed an innovative framework that integrates PCA-GRA, StiePat-MLR, and GM (1,1)-BP neural network models to refer to the world's largest container port, Shanghai Port, and explore drivers related to carbon emissions. The analysis framework proposed in this paper for calculating the port carbon emission factors of Shanghai Port can also be used to measure the carbon emission factors of other ports. The specific conditions of different ports need to be analyzed in detail, and the significance of the conclusions of the factors affecting carbon emissions in each port may have some differences.

The results show that: (1) since 2008, the energy consumption of Shanghai Port has shown an overall growth trend, and the carbon footprint of Shanghai Port Group's energy consumption has shown a slow growth trend. And power consumption is the main source of Shanghai Port's carbon emissions. From 2008 to 2019, under SIPG's implementation of energy-saving and emission-reduction measures, the proportion of Shanghai Port's electricity consumption has increased year by year. However, due to the strong trade demand after 2020, Shanghai Port will increase the proportion of fuel and diesel consumption to meet the surge in shipping demand. (2) The daily operation of the port will generate carbon emissions. After calculation, it is found that the main factors affecting the carbon emissions of Shanghai Port are container throughput, throughput energy consumption, number of berths, net profit attributable to the parent company, and total foreign trade import and export. Among them, the greatest impact on Shanghai Port's carbon emissions is the number of berths; (3) the growth rate of Shanghai Port's carbon emissions will continue to slow down in the next ten years and will reach the carbon peak point around 2033.

Based on the above results, management suggestions for Shanghai Port are put forward, including: (1) Establishing an energy forecasting and early warning mechanism, and an emergency response mechanism to meet the needs of trade fluctuations; (2) Vigorously promoting the construction of green ports and improve energy efficiency; Carry out carbon emission reduction monitoring from the chain point of view, optimize port operation management; and (3) improve berth utilization and reduce loading and unloading time.

**Author Contributions:** Conceptualization, Y.Z. and X.Y.; methodology, Y.Z.; software, B.H.; validation, Y.Z., X.Y. and B.H.; formal analysis, Y.Z.; investigation, B.H.; resources, X.Y.; data curation, Y.Z.; writing—original draft preparation, Y.Z.; writing—review and editing, B.H.; visualization, Y.Z.; supervision, X.Y.; project administration, X.Y.; funding acquisition, X.Y. All authors have read and agreed to the published version of the manuscript.

**Funding:** This research received no external funding.

**Data Availability Statement:** The data presented in this study are available on request from the corresponding author.

**Conflicts of Interest:** The authors declare no conflict of interest.

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
