# Peer review of "Analysis of Carbon Emission Reduction at the Port of Integrated Logistics: The Port of Shanghai Case Study"

_sustainability, doi:10.3390/su151410914_

Round 1
Reviewer 1 Report
The aim of this paper is to deal with the issue of carbon emissions and introduce relevant control policies and low-carbon development strategies to address the problem of controlling carbon emissions, because so far, the academic driving factors that might affect carbon emissions is still insufficient. To tackle this problem, this paper proposes three calculation models or Shanghai Ports carbon emissions that conform an innovative framework that integrates PCA-GRA, StirPat-MLR, and GM (1,1)-BP neural network models to refer to the world's largest container port, Shanghai Port, and explore drivers related to carbon emissions related to carbon emissions factors and predictions.
The importance of this study is to use a systematic combination method to reduce the carbon emissions produced by the port of Shanghai.
English must be proof-read before publication.
All figures must be explained in the manuscript, otherwise, it is difficult to understand why they are shown in the paper.
The relation between their discussion and conclusion with the results is quite weak, I would recommend the authors of the paper to better explain the results to better understand their discussion and conclusions. It is very important to explain the collinearity problem with the regression analysis, what is the consequences of having collinearity between the variables? And how does it affect their framework? If collinearity is not a problem and therefore the regression analysis is not needed, why do you show it as part of your framework?
The aims of the paper and the proposed methodology or framework are very interesting. However, it is a case study, therefore, I would encourage you to discuss, at least, how do you expect this methodology or framework to be applied in other ports? And explain whether you expect the results to remind or change. Or, it would be meaningful if you can test another port.
The english must be proof read, you have few typos and grammatical errors, but in general, it is ok.
Reviewer 2 Report
Dear sirs. It is an interesting and up to date topic, dealing with the carbon emissions in ports. In this case, the port of Shanghai.
Once beginning the paper reading, some points could be improved, for better understanding, as:
- Between lines 51 and 53; figures on carbon emissions in ports, are published, but there seems a confusion in between the size of the figures shown, from 7.795 million tons to 838,33 million tons.
- Line 70 finishes with no point and the sentene seems incomplete.
- SBM model is defined in line 82, but no brief description is made.
Previous introductory and literature review, is focus on port pollution examples and studies. Once entered in methodology aspects, different statistical models are mentioned and data and sequences are explained.
Carbon emission sources are identifed in different areas as logistics, cranes, port transportation vehicles and materials consumed in port. So an exhaustive analysis calculation.
A wide statisctical analysis is done. Seems that the main objective of the paper is the statistical study.
In line 634 a future verbal time is mentioning the year 2020, should be clarified if the year is wrong.
In line 740 Los Angles should be corrected.
Reviewer 3 Report
Please, consider comments and considerations detailed in the attached document with the aim at improving the quality and interest of the paper. Thank so much in advance.

Round 2
Reviewer 3 Report
Thanks for considering all my comments and suggestions.
About the "Response 3", please could you specify what changes have been made in Figure 1 of this updated version versus the previous one?
About the "Response 8", what happens with variables "P3" and "E1"? why are they excluded from the correlation model?
About the "Response 12", could you please mention at least one result of the study which support each of the strategies?
Thanks again for considering these 3 last points from my side.
